# Evaluation of Fecal Coliform Prevalence and Physicochemical Indicators in the Effluent from a Wastewater Treatment Plant in the North-West Province, South Africa

**DOI:** 10.3390/ijerph17176381

**Published:** 2020-09-02

**Authors:** Stenly Makuwa, Matsobane Tlou, Elvis Fosso-Kankeu, Ezekiel Green

**Affiliations:** 1Department of Biotechnology, University of Johannesburg, Doornfontein 2028, South Africa; egreen@uj.ac.za; 2School of Physical and Chemical Sciences at the North-West University, Mafikeng 2790, South Africa; Matsobane.Tlou@nwu.ac.za; 3Water Pollution Monitoring and Remediation Initiatives Research Group, School of Chemical and Minerals, Engineering at the North-West University, Potchefstroom 2520, South Africa; Elvis.FossoKankeu@nwu.ac.za

**Keywords:** influent, effluent, wastewater, *E. coli*, physicochemical, heavy metals, general limits, special limits, green drop

## Abstract

Compliance of the effluents from wastewater treatment plants (WWTPs) to the regulatory standards, which mostly entail the removal/reduction of organic waste and deactivation of the potential microbial pathogens is of great importance. The detection of indicator parameters can be used to determine the effectiveness of a WWTP and the level of compliance with the South African regulatory standards. The performance of the WWTP was assessed by biological, physical and chemical measures in wastewater final effluent. The *Escherichia coli* ranged from 0 and 2420 count/100 mL in the final effluent. The recorded values for the physicochemical parameters were within the following ranges: pH (7.03–8.49), electrical conductivity (81.63–126.5 mS/m), suspended solids (0.40–20.4 mg/L), ammonia (0–22.15 mg/L), Chemical Oxygen Demand (COD) (1–73 mg/L), nitrate (0–16.1 mg/L), ortho-phosphate (0–8.58 mg/L) and free chlorine (0–3.21 mg/L). Furthermore, the concentration of toxic heavy metals was recorded to be between 1–10 ug/L for arsenic, cadmium, lead and mercury. In conclusion, all the parameters that were evaluated in this study indicate that the studied WWTP is performing in accordance with the prescribed general limits.

## 1. Introduction

Wastewater is derived from raw sewage from anthropogenic activities [1,2], while sewage effluent refers to treated or untreated wastewater generated from a treatment plant [3]. The main aim of a municipal treatment plant is to remove/reduce organic wastes and safeguard human health by deactivating disease-causing microorganisms [4]. Insufficient management of municipal wastewater in many urban areas is a major challenge that leads to poor quality effluents entering public surface waters and thus posing high risk to downstream users and aquatic life [2,4,5].

In South Africa (SA), 19% of the rural population lacks access to potable water [6] and therefore, continues to use untreated surface water which is often contaminated by polluted wastewater effluents [7,8]. Although the quality of effluents for most WWTPs has been adequate to meet discharge regulations and standards in the past, the drastic increase in population for a developing country like SA, often means high volumes of wastewater discharged into the environment and consequently, environmental pollutants that are a threat to public health [9,10].

One of the main aspects of wastewater treatment is the removal/reduction of the contaminating microorganisms [11]. The presence of fecal coliforms in final effluents therefore confirms the presence of human pathogens in wastewater and the potential to contaminate natural water/environmental water resources [11,12]. As a result, fecal coliforms are often used as indicators for water quality and can provide valuable information on urban land use and potential routes of fecal contamination [11,12,13,14]. *E. coli* is a preferred indicator organism to monitor the bacteriological quality of wastewater [15]. On the other hand, inadequately treated effluent may contain hazardous elements [16,17,18] which place sufficient treatment technologies/processes [19] at the cornerstone of any WWTP. Some of the hazardous elements are in the form of heavy metals and the most commonly reported include arsenic, lead, mercury and cadmium [20,21] which are known to be carcinogenic and mutagenic [22]. The sources of most physicochemical parameters in wastewater are mainly associated with households, grey water and black water, mines, domestic and industrial activities, humus materials, waste created by animals and other living organisms present in water [20,23]. Some of the chemical parameters that include heavy metals, are persistent to treatment due to their non-biodegradable and toxic nature [24].

Wastewater needs to be adequately treated prior to being discharged into the environment. Thus, the importance of testing for indicator contaminants for quality characterization, control and compliance of wastewater final effluent as the framework of quality programs, is widely important and accepted nowadays [25]. There is a need to improve treatment processes and to adopt stringent policies in terms of monitoring and control of the quality of the final effluent in order for a treatment plant to meet national and international standards [26]. In SA, the compliance of a plant to regulatory standards is recognized and awarded with an incentive-based program certificate popularly known as a green drop certificate [27,28]. The regulatory standards are used to measure the quality of the effluent, while the green drop program is driven to strengthen the regulatory approach, and at the same time refocus the local government in a manner that is more responsive to regulatory imperatives [27].

Studies have been carried on WWTP, but few or no papers have been done in the North-West Province of South Africa about their compliance to general and special limits as well as their performance when measured to current updated green drop effluent quality requirements. The aim of this study is intended in assessing the prevalence and compliance of the studied plant to South African regulatory standards with regard to fecal coliforms and physicochemical parameters as well as to improve knowledge on the quality of wastewater discharged to the environment due to various anthropogenic activities.

## 2. Materials and Method

### 2.1. Study Area

The plant considered in this study is situated in the North-West Province of SA. According to regulation 2834, popularly known as regulation 17, the plant is registered and classified as class A [29]. The town where the plant is situated has a population of around 124,000. It is an industrial and agricultural growth point of the North-West province. The plant therefore receives municipal domestic sewage and wastewater that is heavily influenced by industrial water use.

There are different technologies for treatment of wastewater in SA, of which activated sludge and trickling filter are the commonly applied methods [30]. The plant is an activated sludge with a capacity of 45 megaliters (ML) and is divided into two sections (old and new plant) with operational capacities of 23 and 22 ML, respectively. The old plant is subdivided into polishing and Bardenpho sections with a capacity of 10 and 13 ML, respectively, while the new plant is subdivided into reactor A and B with operational capacity of 11 ML each.

### 2.2. Sample Collections

Both raw and treated wastewater samples were collected aseptically using sterile 250 mL and 1 L sampling bottles respectively for microbiological and physicochemical testing. The sampling containers were washed with soap and water and autoclaved after each use. Samples were collected between May 2019 and March 2020. Physicochemical samples were collected at the plant inlet (raw sewer) and plant outlet (final effluent). Microbiological samples were collected before disinfection (secondary effluent) and after disinfection (final effluent).

### 2.3. Detection of Fecal Coliforms (E. coli)

The Colilert Quanti-Tray/2000 system as described in Omar et al., [31] was used for the enumeration of the viable *E. coli* cells from the 336 samples studied. Enumeration of *E. coli* from samples was done by using 100 mL water according to the manufacturer’s instructions. The Quanti-Trays were incubated for 18–22 h at 37 °C. After incubation, the Quanti-Trays/2000 were examined under long wave (366 nm) ultraviolet light, and wells that turned both yellow and fluoresced were counted as *E. coli* positive (IDDEX). The results of the quantifications were reported as *E. coli* count/100 mL.

### 2.4. Determination of Physicochemical Parameters

All equipment and meters were verified and calibrated according to the manufacturer’s instruction. A total of 336 samples were analyzed for all physicochemical parameters except for heavy metals which were analyzed from 20 samples. Free chlorine was measured onsite using HACH handheld pocket colorimeter Pocket Colorimeter II 5870000 (HACH Company, Loveland, CO, United State of America (USA)). pH was measured using benchtop pH meter (XS Instruments, Via della Meccanica, Italy). Ammonia, nitrate, and ortho-phosphate from raw and treated samples were analyzed using spectrophotometry model DR3900 (HACH Company, Loveland, CO, USA) and a Gallery Discrete Analyzer Thermo Scientific (Thermo Fisher Scientific, Waltham, MA, USA) respectively. COD samples were digested in a Hanna thermo reactor HI 839800 COD reactor (Hanna Instruments, Woonsocket, RI, USA). Electrical conductivity was measured using a benchtop meter Eutech Instruments-Con 2700 (Thermo Fisher Scientific, Waltham, MA, USA). Whatman GF/F Glass Microfiber 47 mm filters (Sigma-Aldrich, St. Louis, MO, USA) were used for the measurement of suspended solids. Heavy metals analyses from the final effluent samples were outsourced to Waterlab (Pty) Ltd. in Pretoria, South Africa. Analyses for the heavy metals were done twice every month through inductively coupled plasma optical emission spectrometry (ICP-OES).

### 2.5. Compliance Study and Calculation of Reduction Efficiencies

The Department of Water Affairs General Authorization guidelines (general and special limits) as indicated in Table 1 [32] along with the green drop category requirements (microbiological, chemical and physical) were used as yardsticks/benchmarks to evaluate the acceptability of the final effluent of the studied plant. General limits are applicable to WWTPs discharging effluents of less than 2 ML as well as discharging into water resources that are not listed on the regulation [32], while special limits apply to WWTPs discharging effluents less than 2 ML, but discharging into a water resource listed in the regulation [32]. However, WWTPs that discharge effluents above 2 ML and do not have a water use license also use these regulation limits. All effluent quality compliance categories (microbiological, chemical, physical compliance) must obtain ≥95%, respectively, for the plant to meet the green drop requirements [33].

The efficiency of the studied plant for the removal of waste matter was calculated using the following equation [4,34]:(1)$$\mathrm{Removal}\;\mathrm{efficiency}\;\left(\%\right)=\frac{\mathrm{Ci}-\mathrm{Ce}}{\mathrm{Ci}}\times 100$$
where Ci is the concentration of waste matter in influent and Ce the concentration of waste matter in effluent.

## 3. Results

Presumptive *E. coli* detected as shown in Table 2, ranged between 0 and 2420 *E. coli* count/100 mL. The highest count of 2420 *E. coli* count/100 mL was observed in November, December and January. The organism was detected in 96.43% of the 336 final effluent samples studied, with 98.81% of the samples below the permissible general limit of 1000 counts/100 mL as recommended by the South African General Authorization regulation in Table 1 [32]. The study results were also evaluated against the special limit and the outcomes indicated a lower compliance of 3.57% out of the 336 samples studied. Zero detection as required by the special limit, was only observed in 12 samples. The final effluent samples with no detection (0 *E. coli* count/100mL) were observed in June, August, December and March, respectively. The final effluent was above the requirement for green drop when evaluated against the general limit. The 3.57% obtained through special limit, was too little to meet the green drop limit. The plant has a chlorination (disinfection) step as the only tertiary treatment before effluent discharge. In terms of monitoring the efficiency of the chlorination system, 99.77% of the organisms were reduced from the 1.26 × 10^5^ count/100mL detected before disinfection. The study recorded *E. coli* reduction of more than 99% in all the months studied (see Table 3 and Appendix A). Few cells of *E. coli* survived disinfection from the secondary effluent samples.

The physicochemical parameters were categorized into physical and chemical parameters (Table 2). The chemical parameters were complemented by heavy metals analyses. The results were compared against SA General Authorization (general and special limits). The concentrations of the physical parameters attained, were ranged as follows: pH (7.03–8.49), electrical conductivity (81.63–126.5 mS/m) and suspended solids (0.40–20.4 mg/L). All the samples tested for physical parameters results complied 100% to the permissible general limit. However, when evaluated against special limits, 104 (30.95%), 71 (21.13%) and 289 (86.01%) compliances were observed for pH, electrical conductivity and suspended solids, respectively. The lowest concentrations for electrical conductivity and suspended solids were observed in September and August, respectively. The physical parameters met the ≥95% requirement for green drop when evaluated against the general limit, while noncompliance of 46.03% was observed when measured against the special limits.

Ammonia, COD, nitrate, ortho-phosphate and free chlorine were studied under the chemical category. Their detected concentrations ranged between 0–22.15 mg/L, 1–73 mg/L, 0–16.1 mg/L, 0–8.58 mg/L and 0–3.21 mg/L, respectively. The final effluents of all the samples exhibited 100% compliance when measured against general limits. Compliances above 90% were observed for ammonia (92.86%) and nitrate (99.7%) while lowest compliance to the general limit was observed with free chlorine (18.75%). The final effluent concentrations when evaluated against the special limits, resulted in the following compliances: 75% for ammonia, 61.31% for COD, 22.13% for nitrate, 98.63% for ortho-phosphate and 0.3% for free chlorine. The efficiencies of the plant in reducing ammonia, COD, nitrate and ortho-phosphate from the inlet concentrations as indicated in Table 3 and Appendix A, were 94.23%, 31.51% and 91.46%, respectively. The challenge around the reduction of nitrate was due to the configuration employed. The treatment process in reactor A had to be changed from Modified University of Cape Town (UCT) to Phoredox, that brought stability in reducing the concentration of nitrate. The change around the configuration from Modified UCT to Phoredox happened in October and the change showed better outcomes. Before the change, the plant would sometimes experience a higher concentration at the final effluent than the raw inlet. The efficiency of the plant to remove COD was improved as the plant managed to reduce the concentration from the inlet of the plant by more than 95%. The lowest reductions that were below 95% but higher than 90% were observed in December, January and March.

Heavy metals such as arsenic, cadmium, mercury and lead were also part of the chemical parameters studied under the chemical category. The concentrations of these heavy metal parameters in ug/L at the final effluent as indicated in Table 2 were as follows: arsenic 1–2 (1.02), cadmium 1–3 (1.45), mercury 1–6 (1.55) and lead 1–10 (3.05). All the samples were 100% compliant for arsenic, cadmium and lead when measured against general limits, while mercury had two failures with a compliance of 90%. Arsenic was the only heavy metal that had 100% compliance to both general and special limits, while a decrease in compliance was observed with cadmium (75%), mercury (85%) and lead (75%) when evaluated against special limits.

To determine the chemical category compliance to the set green drop requirement, the chemical parameters obtained were 89.03% and 65.82% when the effluent was evaluated against general and special limits, respectively. The findings were however lower than the required ≥95% for compliance to the green drop in all the regulatory standards (general and special limits).

## 4. Discussion

*E. coli* has been increasingly used as a fecal coliform indicator due to the fact that it is the only organism amongst other coliforms that is of fecal origin even though some of its strains are not considered pathogenic [35,36]. The organism accounts for more than 95% of the coliform genera in human feces [37]. The *E. coli* counts in 1.91% and 96.43% of the samples tested in this study were over the set General Authorization guidelines of 1000 counts/100 mL (general limit) and 0 count/100 mL (special limit), respectively, and indicated noncompliance in this regard. The results reported here and the issues of compliance that characterized this sector are further demonstrated in a study by Osuolale and Okoh [11], on the compliance of WWTP-B to the general but not the special limit and other reports of a similar nature [38,39]. In this study, the samples that were compliant to special limits, were only observed in June, August, December and March. The detection of *E. coli* in wastewater final effluents presents a major threat to public health [40] and possibly indicates the presence of other disease causing/potentially pathogenic microbes including viruses and protozoa. Osuolale and Okoh’s [11] study identified their findings with insufficient chlorine as the cause of high *E. coli* detections. However, in this study, residual chlorine proved to be high, which therefore confirmed that enough chlorine for the disinfection process was being dosed. Some aspects (cleaning of clarifiers, sludge reticulation affected due to aerator breakdown, high inflow of 103% at balancing dam and load shedding) that are not associated with disinfection have been identified on the days with high detections above general limit. The effluent in this study was less polluted than the effluent in a study by Elmund et al., [41] for which the detections ranged between 0 to approximately 10,000 and 3 to 1500 organisms/100 mL at their two plants studied. About 81.25% of free chlorine meant to eradicate these organisms was above the set limit when measured against the general limit. The final effluent compliance was about 3.57% when measured against zero *E. coli* count/100 mL of the special limit. The high free chlorine concentration detected confirmed that the dosed chlorine was more than sufficient and possibly indicated the presence of chlorine resistant strains [18]. The presence of resistant microorganisms pose significant risk to public health. Studies have shown stress mechanisms [18,42,43] and resistance genes mechanisms [44,45] advancing in the resistance of *E. coli* to disinfection by chlorine. As the final effluent samples, the samples collected prior to disinfection were also analyzed to determine the number of organisms expected to be eradicated. This was done to measure the efficiency of the plant in removing microorganisms such as *E. coli*, and the outcomes showed high efficiency of more than 99% even though the compliances for both limits were not optimally (100%) achieved. A study by Edokpayi et al., [39] showed reduction of *E. coli* to levels between 15% and 25% which tend to be lower than the current study reductions. The paucity of *E. coli* present an optimistic picture of the state of this WWTP.

The SA final effluent discharge guidelines outline the limits for physical and chemical parameters [32]. Studies have reported that physicochemical parameters have synergetic effects on each other, which have some impact on the water quality [23,46,47,48]. The outcome of the physical parameters (pH, electrical conductivity and suspended solids) analyzed was 100% compliance to the general limits. However, a decline in compliance was observed when the parameters were evaluated against the special limits. Suspended solids showed the highest compliance when measured against the special limit. Studies in the Limpopo and Gauteng provinces of SA reported general limit compliance with regards to pH [4,34], as was also observed in this study. Electrical conductivity in this study ranged between 81.63 and 126.5 mS/m. The electrical conductivity from final effluent in the Mpumalanga Province (SA) ranged from 162.1 to 1170 mS/m for the final effluent [26]. Studies done in Eastern Cape by Agoro et al., [4] and Osuolale and Okoh, [11] showed higher detections of electrical conductivity above both South African general and special limits. Suspended solids showed the highest compliance when measured against the special limit. Like other indicator parameters, suspended solids are used to describe the extent of pollution in wastewater [48]. The average of suspended solids recorded in this study were lower compared to three of the four plants studied in Vaal, Gauteng [34].

The data presented here indicates that for the vast majority of the samples, the general limit criteria for discharging chemical parameters were met. These results correspond well with the data obtained from the study in the Eastern Cape Province (EC-SA) [49,50] and Bulgaria [50] which reported on data that is compliant to the general limit but noncompliant to the special limit due to a diversity of reasons [48,50,51]. The 100% compliance reported in this study with respect to COD was in the concentration range between 1–73 mg/L which was significantly higher in comparison to the 33% compliance that was reported in the EC-SA WWTP [11]. Igbinosa and Okoh, [40] recorded COD effluent concentrations in the range of 36.82–238 mg/L from a WWTP in the East of Alice Town of EC-SA.

Chlorine was one of the chemical parameters that were not in compliance to both general and special limits. This parameter is widely used to disinfect wastewater prior to discharge in order to mitigate against the spread of waterborne infectious diseases [52]. Moreover, chlorine can be harmful to aquatic life if high concentrations are discharged into the environment [53]. It is therefore regulated before being discharged into the environment as free or residual chlorine in SA [32]. Only 63 (18.75%) and 1 (0.30%) were compliant to general and special limits. About 67% of the WWTP-A samples in EC-SA had low free chlorine concentration below the recommended general limit [11]. Most of SA WWTPs use chlorine as a form of disinfectant and it is expected to detect free chlorine in the final effluent after contact time of 30 min, however contrary to that, the special limit requires a discharge of 0 mg/L and thus seems to be too stringent a requirement for a plant.

The study further showed the effectiveness of the WWTP in reducing the concentration of ammonia, COD, ortho-phosphate present in the influent except for nitrate. The reduction of these parameters were above 90%. Phosphate which is an essential nutrient for plants [54] was considerably reduced compared to 24% and 5% recorded in a study conducted in Limpopo [39]. Contrary to this study, higher nitrate reduction efficiencies of 98.3%, 85.8%, 96.9%, 86.2% and 75.8% were recorded in a study by Aniyikaiye et al., [55]. The plant employed different activated sludge configurations such as Modified UCT, Bardenpho and Phoredox for reductions of nitrate, ammonia and phosphate [56,57]. The challenge around the reduction of nitrate was due to the configuration employed which sometimes showed higher concentrations of nitrate in the final effluent than at the inlet of the plant. Similar trends (higher nitrate concentration at the final effluent than the inlet), were also observed for the majority of parameters studied by Edokpayi et al., [39]. High concentration of nitrate is known to accelerate algal growth causing eutrophication, which consequently leads to increase in oxygen demand, loss of some aquatic life forms and offensive odors that affect people living very close to the water resource [48,58,59,60,61]. The issue of noncompliance with regard to nitrate was a concern and it led to the changes in configuration applied from Modified UCT to Phoredox which showed efficient outcomes. Most activated sludge configurations have advanced over the years, and, subject to the design, an activated sludge WWTP can achieve removal of organic carbon substances and nutrients such as nitrogen and phosphorus [56]. The outcomes of this study show the efficiency of the activated sludge configurations applied except for the Modified UCT configuration.

The plant considered in this study is situated in an area that is known for intense industrial activity in the province. Different industrial wastewater streams from the industrial catchment area are mixed with domestic sewage discharges before they reach the treatment system. Effluents from industry are mostly dominated by heavy metals that are persistent to treatment and are toxic to the environment [24]. The study of the chemical properties from the final effluent was completed by analyzing the heavy metals. The studied heavy metals are shown in Table 2. The plant showed high efficiency in reducing heavy metals resulting in a final effluent compliance ≥90% and ≥75% when evaluated against general and special limits, respectively. Arsenic concentrations were above the general and special limits, while cadmium concentrations complied with both limits in three plants studied in Gauteng [34]. The detection range for cadmium and mercury were lower in a Cape Town study by Olujimi et al., [62], compared to the detection limit in this study. According to the literature, high trace levels of arsenic, cadmium and mercury were recorded in Austria [63], Italy [64] and Israel [65]. A study by Edokpayi et al., [39], showed the concentration of lead in the effluent complying with the DWA guideline value of 10 ug/L [25] except for months of April and June. Similar findings were observed in this study with lead complying 100% to both general and special limits. A study in China indicated cases of lead above the Chinese standard and such occurrence was mostly observed from WWTPs receiving wastewater from electroplate plants [66].

The green drop process measures the performance of a WWTP, and subsequently rewards the institution upon evidence of their excellence according to the minimum standards or requirements that have been defined. Since the data was evaluated through General Authorization (general and special limits), the plant classified as B according to SA classification of WWTP is then expected to meet ≥95% for compliance to green drop [32,33]. Overall achievement around microbiological, chemical and physical green drop compliance in relation to general and special limits as indicated in Figure 1 was as follows: general limits (microbiological—98.1%, Chemical—89.03% and Physical—100%) and special limits (microbiological—3.57%, Chemical—65.82% and Physical—46.03%). Compliance of the plant in relation to general limits had two categories (microbiological and physical) above the set target of ≥95%. Chemical determinant was the only category that was not in compliance. This therefore would not award the plant green drop certification status because of the fact that not all categories achieved ≥95% compliance. The results showed overall noncompliance to green drop when measured using special limits. Contrary to the above findings, the plant had achieved its green drop status for several years since its inception in 2008, except in 2009 [28]. A decline was observed in the performance of the plant when compared to the last audit that was done in 2013. In that year (2013) the plant obtained 99.71%, 99.93%, and 100% for microbiological, chemical and physical categories, respectively [28]. Despite the decline in compliance, the conditions for all the categories were initially set at ≥90% before being reviewed to ≥95% [33]. This therefore indicates that the harsher the green drop compliance requirements become, the less likely it is that the plant will obtain its green drop status, unless the design of the plant is revised as the infrastructure is ageing. Insufficient budget for assets management and ageing infrastructure have been identified as the source of decline for the performance of the plant.

## 5. Conclusions

Microbiological and physical treatments are very significant within wastewater treatment systems. Wastewater with high domestic content has the highest negative impact on water quality in a river. The plant of study showed high efficiency in reducing sewer constituents. The analytical outcomes confirmed compliance of the final effluent to general limits, however, some challenges were observed for compliance to special limits. Detection of high residual chlorine concentrations were observed, therefore there is a need for introspection on the issue of chlorine dosing because higher residual concentrations that are not compliant to both SA standards are being discharged into the environment and that is not good for aquatic life. Discharging effluents containing *E. coli* and high residual chlorine imply chlorine as the only source of disinfection is not sufficient; therefore, complementing the chlorine disinfection process at the current plant with maturation ponds, ozonation, or ultraviolet radiation could radically assist the plant in discharging effluents with zero *E. coli* which would illustrate an absence of pathogens. The Phoredox configuration proved to be efficient in reducing nitrates compared to Modified UCT. Even though challenges have been observed with compliance to special limits and green drop requirements, the plant however showed high efficiency in reducing majority determinants in line with general limits standards.

## Figures and Tables

**Figure 1 ijerph-17-06381-f001:**
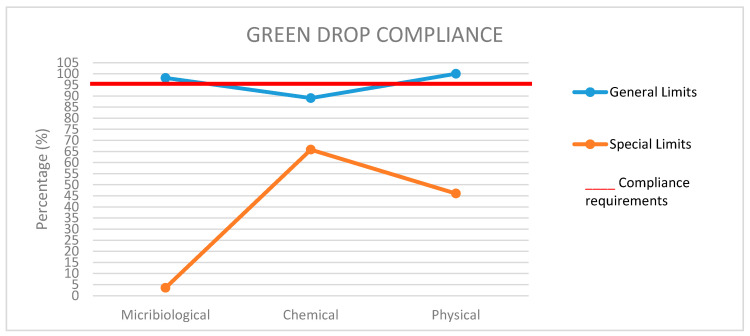
Assessment of the data to green drop compliance.

**Table 1 ijerph-17-06381-t001:** South African General Authorizations for general and special limits [32].

Substance/Parameter	General Authorizations	
	General Limits	Special Limits
Fecal coliforms (per 100 mL)	1000	0
Chemical oxygen demand (COD) (mg/L)	75 (i)	30 (i)
pH	5.5–9.5	5.5–7.5
Ammonia (as N) (mg/L)	6	2
Nitrate (as N) (mg/L)	15	1.5
Chlorine as free chlorine (mg/L)	0.25	0
Suspended solids (mg/L)	25	10
Electrical conductivity (mS/m)	75 mS/m above intake water, to maximum of 150	50 mS/m above background receiving water, to maximum of 100
Ortho-phosphate as phosphorous (mg/L)	10	1 (medium) and 2.5 (maximum)
Fluoride mg/L	1	1
Soap, oil or grease (ug/L)	2500	0
Dissolved arsenic (ug/L)	20	10
Dissolved cadmium (ug/L)	5	1
Dissolved chromium (ug/L)	50	20
Dissolved copper (ug/L)	10	2
Dissolved cyanide (ug/L)	20	10
Dissolved iron (ug/L)	300	300
Dissolved lead (ug/L)	10	6
Dissolved manganese (ug/L)	100	100
Mercury and its compounds (ug/L)	5	1
Dissolved selenium (ug/L)	20	20
Dissolved zinc (ug/L)	100	40
Boron (ug/L)	1000	500

**Table 2 ijerph-17-06381-t002:** Physical parameters compliance to South African General Authorizations regulations.

Categories	Parameter	Unit	Mean	Min	Max	* Std. dev.	General Limits		Special Limits	
							Results >Limit (n)	** Cpl. (%)	Results >Limit (n)	** Cpl. (%)
Microbiological	*E. coli*	Count	69	0	2420	249.09	4	98.81	324	3.57
		/100 mL								
Physical	pH	-	7.71	7.03	8.49	0.3	0	100	232	30.95
	Electrical conductivity	mS/m	108.23	81.63	126.5	8.15	0	100	265	21.13
	Suspended solids	mg/L	7.62	0.4	20.4	3.06	0	100	47	86.01
Chemical	Ammonia	mg/L	2.1	0	22.15	3.63	24	92.86	84	75
	*** COD	mg/L	25.48	1	73	11.86	0	100	130	61.31
	Nitrate	mg/L	5.53	0	16.1	3.96	1	99.7	285	22.13
	Ortho-phosphate	mg/L	0.42	0	8.58	0.62	0	100	5	98.63
	Residual chlorine	mg/L	0.69	0	3.21	0.51	273	18.75	335	0.3
	Arsenic	ug/L	1.05	1	2	0.22	0	100	0	100
	Cadmium	ug/L	1.5	1	3	0.89	0	100	5	75
	Lead	ug/L	1.6	1	6	1.57	2	90	3	85
	Mercury	ug/L	3.25	1	10	4	0	100	5	75

* Std. dev. = Standard deviation; ** Cpl. = Compliance; *** COD = Chemical Oxygen Demand.

**Table 3 ijerph-17-06381-t003:** Wastewater treatment plant (WWTP) removal efficiency in reducing *E. coli*, ammonia, COD, nitrate and ortho-phosphate from untreated wastewater.

Months	*E. coli* (Count/100 mL)			Ammonia (mg/L)			COD (mg/L)			Nitrate (mg/L)			Ortho-Phosphate (mg/L)		
	Secondary Effluent	* F.E	** Red %	Raw Inlet	* F.E	** Red %	Raw Inlet	* F.E	** Red %	Raw Inlet	* F.E	** Red %	Raw Inlet	* F.E	** Red %
May-19	4.93 × 10^4^	70	99.86	43.64	0.25	99.43	674	27	95.99	9.24	7.66	17.1	5.47	0.49	91.04
Jun-19	2.06 × 10^4^	46	99.78	39.81	0.57	98.57	694	32	95.45	8.29	8.52	-2.77	4.97	0.61	87.73
Jul-19	1.03 × 10^4^	36	99.65	41.24	0.2	99.52	709	27	96.13	10.6	11.17	-5.38	5.62	0.38	93.24
Aug-19	1.39 × 10^4^	35	99.74	51.98	0.22	99.58	727	29	96.08	7.93	9.75	-22.95	6.12	0.36	94.12
Sep-19	7.45 × 10^3^	68	99.09	39.21	0.09	99.77	792	33	95.83	8.65	8.94	-3.35	6.92	0.48	93.06
Oct-19	1.17 × 10^5^	54	99.95	40.46	0.99	97.55	805	28	96.55	11.82	4.71	60.15	7.13	0.35	95.09
Nov-19	5.42 × 10^5^	103	99.98	49.58	7.76	84.35	741	27	96.32	10.19	1.19	88.32	7.31	0.22	96.99
Dec-19	3.07 × 10^5^	98	99.97	27.64	5.56	79.88	413	24	94.13	6.19	1.97	68.17	4.27	0.32	92.51
Jan-20	5.67 × 10^4^	123	99.78	28.06	0.89	96.83	206	15	92.56	3.35	2.28	31.94	3.36	0.45	86.61
Feb-20	2.29 × 10^4^	76	99.67	45.04	2.91	93.54	356	17	95.18	4.75	1.66	65.05	4.3	0.51	88.14
Mar-20	2.42 × 10^5^	51	99.98	31.75	3.97	87.5	293	21	92.89	5.23	2.6	50.29	3.69	0.46	87.53
*** Ave	1.26 × 10^5^	69	99.77	39.86	2.13	94.23	583	25	95.19	7.84	5.5	31.51	5.38	0.42	91.46

* F.E = final effluent; ** Red % = reduction percentage; *** Ave = average.

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
