# Peer review of "Evaluation of Fecal Coliform Prevalence and Physicochemical Indicators in the Effluent from a Wastewater Treatment Plant in the North-West Province, South Africa"

_ijerph, 2020, doi:10.3390/ijerph17176381_

Round 1
Reviewer 1 Report
The manuscript is focused on the quantification of biological, physical and chemical parameters in raw and treated wastewater samples in order to evaluate the performance of a treatment plant in South Africa.
In my opinion, the paper is well written and organized. The results are promising and discussed in detail. However, there are some points that should be revised:
- Line 32: References are not correctly written. They should appear as [1,2] instead of [1],[2]. Please, revise this aspect through all the manuscript.
- Line 35: I think it should say: “deactivating disease caused by microorganisms”.
- Line 81: It would be better if the authors write “a population of around 124,000”.
- Lines 86-89: the font is different as the font of the rest of the manuscript.
- Line 91: Authors said they used sterile sampling bottles. Did they follow a special treatment/method to sterilize?
- Line 114: It should read as follows: “filters 47 mm were used”.
- Line 115: Section 2.4 would be more thorough if the authors indicate which method the Waterlab in Pretoria used to analyse heavy metals.
- Line 118: Could you clarify what the differences or the meanings of General and Special limits are?
- Line 121: it is not clear what authors want to say about the 95% to meet the Green Drop requirement. 95% of what? What’s the meaning of that value?
- Line 133. The section 3 talks about the results obtained in the samples, but authors do not clarify if they are talking about all the samples (including raw and final effluent) or only the final effluent. If they are talking about all the samples, it should be better to separate the data in raw and final effluent, so that the readers can understand better which sample of each category are complying the limits.
- Table 2: There is a line drawn under the row of Microbiological, that should be removed. Also, the words Microbiological, E. Coli, Counts, 69, 0 and 2420 are typed in bold. They should not be in bold. In addition, I think the word Physical should be moved up, so that it is in the same row as the pH, and the word Chemical should be written in the same row as NH3. Please change CD by Cd. And please clarify what CL means. All acronyms should be explained before their use, either in the table or in the manuscript.
- Line 162: It should read: 8.58 mg/L
- Table 3: Please, clarify what F.E means. I guess it is Final Effluent, but it should be clarified.
- Line 272: It should read: “people living very close to”.
Author Response
Comments and Suggestions for Authors
The manuscript is focused on the quantification of biological, physical and chemical parameters in raw and treated wastewater samples in order to evaluate the performance of a treatment plant in South Africa.
In my opinion, the paper is well written and organized. The results are promising and discussed in detail. However, there are some points that should be revised:
1. COMMENT: Line 32: References are not correctly written. They should appear as [1,2] instead of [1],[2].
RESPONSE: The authors appreciates the corrections and suggestions raised by the reviewers, the referencing style has been revised accordingly
2. COMMENT: Please, revise this aspect through all the manuscript.
RESPONSE: The authors appreciates the corrections and suggestions raised by the reviewers, the references has been revised accordingly
3. COMMENT: Line 35: I think it should say: “deactivating disease caused by microorganisms”.
RESPONSE: We do not agree with the reviewer here. WWTP deactivate disease causing microbes/pathogens and not disease caused by microbes.
4. COMMENT: Line 81: It would be better if the authors write “a population of around 124,000”.
RESPONSE: The authors appreciates the suggestions raised by the reviewer, line 78, under the methodology, “a population of around 124 000” was corrected as suggested.
5. COMMENT: Lines 86-89: the font is different as the font of the rest of the manuscript.
RESPONSE: The authors appreciates the corrections raised by the reviewer, the font has been has been changed to match the rest of the manuscript.
6. COMMENT: Line 91: Authors said they used sterile sampling bottles. Did they follow a special treatment/method to sterilize?
RESPONSE: The sentence “The sampling containers were washed with soap and water and autoclaved after each use” in line 90-91 explain the method used for sterilization.
7. COMMENT: Line 114: It should read as follows: “filters 47 mm were used”.
RESPONSE: Line 112-113 was changed to read “Whatman GF / F Glass Microfiber 47 mm filters were used for the measurement of suspended solids.
8. COMMENT: Line 115: Section 2.4 would be more thorough if the authors indicate which method the Waterlab in Pretoria used to analyse heavy metals.
RESPONSE: Line 114-115, the sentence “Analyses for the heavy metals were done twice every month through Inductively Coupled Plasma Optical Emission Spectrometry (ICP-OES)” was inserted to answer the reviewer’s comment.
9. COMMENT: Line 118: Could you clarify what the differences or the meanings of General and Special limits are?
RESPONSE: Line 120-124, explains the differences requested by the reviewer.
10. COMMENT: Line 121: it is not clear what authors want to say about the 95% to meet the Green Drop requirement. 95% of what? What’s the meaning of that value?
RESPONSE: Line 124-126: The 95% mentioned is for the effluent quality compliance that takes into account, microorganisms, chemical and physical compliance. Therefore a sentence “All effluent quality compliance categories (microbiological, chemical, physical compliance) must obtain ≥95% respectively for the plant to meet the Green Drop requirements” to explain the above comment has been inserted.
11. COMMENT: Line 133. The section 3 talks about the results obtained in the samples, but authors do not clarify if they are talking about all the samples (including raw and final effluent) or only the final effluent. If they are talking about all the samples, it should be better to separate the data in raw and final effluent, so that the readers can understand better which sample of each category are complying the limits.
RESPONSE: The results explain the final effluent compared to general and Green drop limits.
12. Table 2: There is a line drawn under the row of Microbiological that should be removed. Also, the words Microbiological, E. coli, Counts, 69, 0 and 2420 are typed in bold. They should not be in bold. In addition, I think the word Physical should be moved up, so that it is in the same row as the pH, and the word Chemical should be written in the same row as NH3. Please change CD by Cd. And please clarify what CL means. All acronyms should be explained before their use, either in the table or in the manuscript.
RESPONSE: All the suggested corrections were made.
13. COMMENT: Line 162: It should read: 8.58 mg/L
RESPONSE: Line 165, the changes were made as suggested.
14. COMMENT: Table 3: Please, clarify what F.E means. I guess it is Final Effluent, but it should be clarified.
RESPONSE: F.E was clarified as a legend underneath the table.
15. COMMENT: Line 272: It should read: “people living very close to”.
RESPONSE: Line 275, the sentence was changed to the suggested, “people living very close to”.

Reviewer 2 Report
This article should be better to improve the information in the different sections, particularly the Materials and Method and Results and Discussion sections and Conclusion need to be explained well with enough support data.
The following comments will illustrate the previous sentence:
Line 78-89, line 251-252 and line 262-270 have mixing fonts, check the font style. Check line 81 for writing in good grammar and syntax.
In Table 2, the data should be presented and calculated their standard deviations.
It needs to be restructured; the distribution of its headers is confused. It needs footnotes describing the abbreviations used on it as well.
Line 144-145 Please include the unit of this number 1.26 x 105.
In Table 3, the data should be presented and calculated their standard deviations.
It is necessary to include footnotes for describing the abbreviations used on the table. A table must be explained by itself, please make a more complete description of this table in the title. It is better if you put the units along with every specific parameter instead of a general unit.
Quantity of data presented is not sufficient because only 4 of the illustrations and tables are presented and all of them need to modification as discussed above. It needs to improve the analysis or statistic application of the experimental data for comparing and interpreting their relationship of evaluation parameter with the environment or health effect. Therefore, presented data could not support to interpretation and conclusion following the objective. Moreover, the improvement or recommendation should be included at the final.
Author Response
Comments and Suggestions for Authors
This article should be better to improve the information in the different sections, particularly the Materials and Method and Results and Discussion sections and Conclusion need to be explained well with enough support data.
The following comments will illustrate the previous sentence:
COMMENT: Line 78-89, line 251-252 and line 262-270 have mixing fonts, check the font style. Check line 81 for writing in good grammar and syntax.
RESPONSE: The suggested corrections were effected.
COMMENT: In Table 2, the data should be presented and calculated their standard deviations. It needs to be restructured; the distribution of its headers is confused. It needs footnotes describing the abbreviations used on it as well.
RESPONSE: Standard deviations were calculated. The heading has been changed and footnote describing the abbreviations used explained below the table.
COMMENT: Line 144-145 Please include the unit of this number 1.26 x 105.
RESPONSE: Line 148: The unit for the value has been included in line 169 and reads as follows, 1.26 x 105 count/100mL
COMMENT: In Table 3, the data should be presented and calculated their standard deviations.
RESPONSE: Another table has been submitted as supplementary item that contains standard deviations.
COMMENT: It is necessary to include footnotes for describing the abbreviations used on the table. A table must be explained by itself, please make a more complete description of this table in the title. It is better if you put the units along with every specific parameter instead of a general unit.
RESPONSE: Line 192-193: The table’s title was changed to “WWTP removal efficiency in reducing E. coli, ammonia, COD, nitrate and ortho-phosphate from untreated wastewater”, units are written in the sub-titles on the table while the abbreviated words are written in full at the bottom of the table.
COMMENT: Quantity of data presented is not sufficient because only 4 of the illustrations and tables are presented and all of them need to modification as discussed above. It needs to improve the analysis or statistic application of the experimental data for comparing and interpreting their relationship of evaluation parameter with the environment or health effect. Therefore, presented data could not support to interpretation and conclusion following the objective. Moreover, the improvement or recommendation should be included at the final.
RESPONSE: The suggested changes were effected in line with the reviewer’s suggestions.

Reviewer 3 Report
This work assessed the prevalence and compliance of the studied plant to South African regulatory standards with regard to fecal coliforms and physicochemical parameters as well as to improve knowledge on the quality of wastewater discharged to the environment due to various anthropogenic activities. It is an interesting content, but arranged structure needs to be further improved. Therefore, it needs minor revision before it is published in this journal. The following issues should be carefully addressed.
- There are some grammatical errors and the sentences are monotonous in the manuscript, the authors need to go through the entire manuscript sentence by sentence;
- Abstract should be improved since it can not adequately attract the readers in this area;
- Authors should clearly mention the novelty of the study in Introduction;
- The figures quality needs to be improved.
- Give more detail analysis and discussion in the section of “Discussion”, and it should be amended by including a brief analysis respect to the use of the reported results to other scale;
- Conclusions should be rearranged.
Author Response
Comments and Suggestions for Authors
This work assessed the prevalence and compliance of the studied plant to South African regulatory standards with regard to fecal coliforms and physicochemical parameters as well as to improve knowledge on the quality of wastewater discharged to the environment due to various anthropogenic activities. It is an interesting content, but arranged structure needs to be further improved. Therefore, it needs minor revision before it is published in this journal. The following issues should be carefully addressed.
1. COMMENT: There are some grammatical errors and the sentences are monotonous in the manuscript, the authors need to go through the entire manuscript sentence by sentence;
RESPONSE: The manuscript was improved as a whole without changing the meaning of the sentences.
2. COMMENT: Abstract should be improved since it cannot adequately attract the readers in this area;
RESPONSE: The abstract has been re-written as suggested.
3. COMMENT: Authors should clearly mention the novelty of the study in Introduction;
RESPONSE: The novelty of the study is addressed on the aim of the study at the end of the introduction. It is a case study where the sewerage receive water (flow through a dolomite rock system) with abnormally high level of hardness, requiring specific adjustment of the disinfection system.
4. COMMENT: The figures quality needs to be improved.
RESPONSE: Figures are improved.
5. COMMENT: Give more detail analysis and discussion in the section of “Discussion”, and it should be amended by including a brief analysis respect to the use of the reported results to other scale;
RESPONSE: The authors thank the reviewer for the suggestion. The discussion has been revised. We hope it meets the reviewer’s satisfaction.
6. COMMENT: Conclusions should be rearranged.
RESPONSE: The authors thank the reviewer for this suggestion. The conclusions have been rearranged.

Round 2
Reviewer 2 Report
Authors have revised your paper following the suggestion of reviewers.